# Adsorption Characteristics of Gas Molecules Adsorbed on Graphene Doped with Mn: A First Principle Study

**DOI:** 10.3390/molecules27072315

**Published:** 2022-04-02

**Authors:** Tingyue Xie, Ping Wang, Cuifeng Tian, Guozheng Zhao, Jianfeng Jia, Chaozheng He, Chenxu Zhao, Haishun Wu

**Affiliations:** 1Key Laboratory of Magnetic Molecules and Magnetic Information Materials of Ministry of Education, School of Chemistry and Materials Science, Shanxi Normal University, Taiyuan 030006, China; tingyuexie@126.com (T.X.); zhaoguozheng@sxnu.edu.cn (G.Z.); jiajf@dns.sxnu.edu.cn (J.J.); 2School of Physical and Electronics Science, Shanxi Datong University, Datong 037009, China; wangping061226@aliyun.com (P.W.); cftian_050@sxdtdx.edu.cn (C.T.); 3Institute of Environmental and Energy Catalysis, School of Materials Science and Chemical Engineering, Xi’an Technological University, Xi’an 710021, China; hecz2019@xatu.edu.cn

**Keywords:** density functional theory (DFT), electronic structure, charge transfer, -pCOHP, Fermi softness

## Abstract

Herein, the adsorption characteristics of graphene substrates modified through a combined single manganese atom with a vacancy or four nitrogen to CH_2_O, H_2_S and HCN, are thoroughly investigated via the density functional theory (DFT) method. The adsorption structural, electronic structures, magnetic properties and adsorption energies of the adsorption system have been completely analyzed. It is found that the adsorption activity of a single vacancy graphene-embedded Mn atom (MnSV-GN) is the largest in the three graphene supports. The adsorption energies have a good correlation with the integrated projected crystal overlap Hamilton population (-IpCOHP) and Fermi softness. The rising height of the Mn atom and Fermi softness could well describe the adsorption activity of the Mn-modified graphene catalyst. Moreover, the projected crystal overlap Hamilton population (-pCOHP) curves were studied and they can be used as the descriptors of the magnetic field. These results can provide guidance for the development and design of graphene-based single-atom catalysts, especially for the support effect.

## 1. Introduction

Hydrogen sulfide (H_2_S) is a toxic and corrosive gas, which can be easily found in raw natural gas, the by-product gas flow of petroleum processing plants (due to the presence of sulfur in crude oil), and the waste gas flow from petrochemical plants, the paper industry, and coal gasification furnaces [1]. Titanium-modified carbonaceous materials can effectively extract hydrogen from hydrogen sulfide [2]. Formaldehyde (CH_2_O) is widely used in building and decoration materials, and is the most common indoor air pollutant [3]. Boron nitride nanosheets with N vacancy defects are treated as a promising candidate for the detection of CH_2_O in pioneering works [4]. Moreover, hydrogen cyanide (HCN), a kind of cyanide, is a very powerful colorless poison. It can inhibit the oxygen consumption of human tissues and is highly lethal to humans and animals [5]. The reactivity and electronic sensitivity of original and structure-modified BC_2_N nanotubes (BC_2_NNT) for the capture of HCN molecules were investigated [6]. These studies show that it is necessary to find effective adsorbents for these high reactive gas molecules.

Graphene-based devices, including photovoltaics [7], spintronic devices [8], gas sensors [9] and graphene-based composite materials [10], have attracted wide attention in the field of two-dimensional materials owing to their excellent electronic, mechanical, photonic and thermal characters [11,12,13,14,15,16,17,18,19,20,21,22]. In addition, graphene has been proved to be a promising carrier for single-atom catalysts (SACs) due to its orbital hybridization and charge transfer between the substrate and single atoms [23,24]. The single Co active site on the graphene base embodies good electrochemical stability and high electrocatalytic activity for the reduction of triiodide [25]. Gold atom embedded in the single hole of graphene has low cost and high catalytic activity for CO oxidation [26]. High catalytic activity and selectivity have become vital aims to be achieved in the areas of chemicals and energy.

Introducing appropriate dopants into graphene materials can significantly improve the chemical activity and stability of the materials. In recent years, theoretical studies have been carried out on the adsorption of NO_2_, NH_3_, SO_3_, H_2_S, HCN and other gas molecules on graphene (GN) doped with metal atoms (TM). These results show that these catalysts have high stability and play a potential role in the field of electronic devices and gas sensors [27,28,29,30,31,32,33]. MnN_4_-GN also has excellent selectivity and sensitivity for CO detection [34]. Al-doped graphene can enhance high sensitivity for detecting CH_2_O molecules [35]. From the above studies, it can be inferred that it is necessary to synthesize precisely modified SACs with a single atom tightly fixed on the support.

It is necessary to consider the selection of a single TM atom catalyst on the support in order to fill this gap. Recently, Robertson et al. found that the Fe-doped graphene has good selectivity of a magnetic and non-magnetic state and the structure of the complex is also stable [36]. Graphene with pyridine nitrogen defects (FeN_4_-GN and MnN_4_-GN) have been successfully synthesized and their electronic properties and formation energies have been changed [37,38]. As an efficient catalyst for CO oxidation, FeN_4_-GN features a stable configuration, low cost and high activity [39]. The catalytic activity of MnN_4_-GN has also been focused on for ORR application [40]. The types of structural defects in graphene include single vacancies (SV) and divacancy (DV), which can dramatically regulate their charge and reactivity properties [41,42]. In addition, the doping of transition metal (TM) atoms can solve the chemical inertness issue of intrinsic graphene, and transition metal atoms can be treated as highly active centers for molecular adsorption. Therefore, we investigated three types of graphene-based catalysts with single-atom manganese. They were expressed as MnX-GN (X = SV, DV or N_4_) systems, respectively. In this work, it was necessary to investigate the electronic structure, geometric stability, adsorption energy, charge transfer, spin polarization and magnetic properties of CH_2_O, H_2_S and HCN gases on MnX-GN in detail by DFT calculation. We further investigated the influence factors of adsorption energy by plotting the predicted density of states (PDOS), -pCOHP and the electron density difference. Finally, the correlation between the adsorption activity and the graphene-based support was further understood by analyzing the electronic properties and Fermi softness of MnX-GN catalysts. This investigation provided here shows that non noble metal-doped graphene supports for the synthesis of high activity SACs are of great significance.

## 2. Computational Details and Methods

All spin polarized density functional theory (DFT) calculations and the projector augmented wave (PAW) pseudopotentials method are performed within the Vienna ab initio simulation package (VASP) [43,44,45]. The generalized gradient approximation (GGA) with the Perdew–Burke–Ernzerhof (PBE) functional is used to describe the exchange and correlation potential [44]. The van der Waals (vdW) interaction between gas and surface of MnX-GN catalysts plays a key role in the adsorption; thus, gas adsorption on metal-doped graphene has been successfully studied with the vdW correction [46,47]. Therefore, the interaction of vdW is considered with DFT-3 in our work [48]. We use a hexagonal supercell (4 × 4 × 1) with periodic boundary condition as the infinite graphene sheet and the supercell vacuum spacing is set to 15 Å, which is sufficiently large to obtain reliable results [49]. The C–C bond length calculated after optimization is 1.426 Å, which agrees well with previous reports [50,51]. All molecule structures can be visualized with the VESTA 3 program [52]. The crystal orbital Hamilton population (COHP) is used to analyze the interaction between molecules and surfaces [53,54,55].

A plane wave energy cutoff was chosen to be 500 eV, and the width of Gaussian broadening scheme is 0.05 eV occupation of electron level. In order to improve calculation accuracy and save time simultaneously, a k-point grid with 7 × 7 × 1 as the center is used to sample the Brillouin region for structural optimization. When the energy change is less than 10 meV/atom, the k-point grid test is finished. The k-point mesh of 15 × 15 × 1 is used to calculate the density of state (DOS). In the geometric optimization process, the position relaxation of all atoms is calculated by conjugate gradient method until the maximum force is smaller than 0.02 eV/Å. The total energy convergence accuracy is 10^−5^ eV for the electronic self-consistent steps. The cubic cells of 15 Å × 15 Å × 15 Å are used to simulate the energy of isolated atoms (or gas molecules). The atomic charge and electron transfer in the adsorption system were evaluated by Bader charge analysis [56].

In order to easily describe the energy and the electron density difference in this paper, the binding energy (*E*_b_) between the adsorbed atoms Mn and X-GN is defined as $E_{b}=E_{\text{MnX-GN}}-E_{\text{X-GN}}-E_{\mathrm{Mn}}$, where *E*_Mn_, *E*_MnX-GN_ and *E*_X-GN_ are the energy of Mn atom in vacuum, the total energy of MnX-GN and the energy of X-GN (X-GN stand for graphene of SV, DV or N_4_), respectively. The adsorption energy ($E_{\mathrm{ad}}$) of gas molecules on MnX-GN is calculated using the expression as $E_{\mathrm{ad}}=E_{\text{gas-Mn}/\mathrm{GN}}-E_{\mathrm{Mn}/\mathrm{GN}}-E_{\mathrm{gas}}$, where *E*_gas-Mn/GN_ indicates the total energies of gases/MnX-GN complex, *E*_Mn/GN_ denotes the total energy of MnX-GN, *E*_gas_ is the energy of gas molecule. A larger negative value of *E*_b_ and *E*_ads_ demonstrates a stronger interaction. The expression defines the electron density difference as $\Delta \rho =\rho_{\text{gas-Mn}/\mathrm{GN}}-\rho_{\mathrm{Mn}/\mathrm{GN}}-\rho_{\mathrm{gas}}$, where $\rho_{\mathrm{Mn}/\mathrm{GN}}$ represents the electron density of MnX-GN, $\rho_{\text{gas-Mn}/\mathrm{GN}}$ is the electron density of gases/MnX-GN and $\rho_{\mathrm{gas}}$ is the electron density of gas.

## 3. Results and Discussion

### 3.1. Model of MnX-GN

The stable geometric configurations of MnSV-GN, MnDV-GN and MnN_4_-GN are shown in Figure 1. Through the calculation, the absorption height, charge transfer, binding energy and the total magnetic moments are summarized in Appendix A. For MnX-GN substrates, three adsorption sites are considered, including single vacancy, divacancy and four nitrogen decorated defects. The binding energies of embedded Mn are calculated with the values of −6.12, −5.86 and −6.61 eV, respectively. The corresponding adsorption heights are 1.40, 0.70 and 0.03 Å, respectively. Estimated from the Bader charge analysis, the Mn atoms are all positively charged with 0.88, 1.06 and 1.28 |e|, respectively, which agrees with previous works [49,57]. For MnX-GN systems, the cohesive energy of Mn bulk (−2.92 eV per atom) is smaller than the binding energy of Mn [58], indicating that the Mn atom can be well scattered in graphite bases. In other words, this kind of substrate can prohibit the aggregation of manganese atoms from clustering. Therefore, our designed systems possess great geometric stability.

The electronic properties, including charge transfer and magnetic moment, of MnSV-GN, MnDV-GN and MnN_4_-GN are shown in Appendix A. The negative value of the transferred charge from Mn to the substrate indicates that Mn acts as the donor of electrons and graphene acts as the acceptor. The total magnetic moments of doped Mn are 3.00, 2.94 and 3.05 μ_B_, respectively, which is consistent with the data in pioneering works [34,57]. As shown in the PDOS of Appendix A, high localized spin density is observed on TM and coordinated nitrogen (carbon) atoms.

### 3.2. Adsorption Structure, Adsorption Energy

The adsorption structures of CH_2_O, H_2_S and HCN on three MnX-GN catalysts were optimized and the most stable structures for each gas were found. The optimized configurations and parameters of adsorption systems are shown in Figure 1 and Figure 2.

In order to investigate the influence of gas adsorption on MnX-GN catalysts, adsorption energies of three gas/MnX-GN catalysts are calculated, as listed in Appendix A. The adsorption energies (*E*_ad_) of CH_2_O on AlN and InN monolayers were −1.04 and −1.05 eV, respectively [59], which is much lower than that on MnSV-GN (−1.86 eV, see Appendix A), indicating that the adsorption activity of MnSV-GN is higher compared to AlN and InN monolayers. The *E*_ad_ of H_2_S on the MnSV-GN surface was −0.83 eV (see Appendix A), which was similar to the NiSV-GN system (−0.70 eV) [60]. The *E*_ad_ of HCN on CoSV-GN is −0.30 eV [61], which is obviously smaller than that on MnSV-GN (−1.11 eV in Appendix A). The *E*_ad_ of HCN on MnSV-GN (−0.60 eV) [29] is obviously smaller than the results in this work (−1.11 eV in Appendix A). Therefore, it is shown that the adsorption activity of Mn is higher than that of Co and Ni systems. In addition, the adsorption energies of gases on the MnSV-GN catalyst were higher than that on MnDV-GN and MnN_4_-GN, corresponding to a higher reactivity. The adsorption energies of gases on MnSV-GN range from −0.83 to −1.86 eV, indicating a chemical adsorption (>0.5 eV) behavior (see Appendix A). Moreover, the chemisorbed behavior can also be reflected in Figure 2, where the molecules can form chemical bonds with Mn atoms in the substrates.

After the gas adsorbing on the MnX-GN catalysts, the uplift heights of Mn atoms are summarized in Appendix A. Gas molecules can lift Mn atoms higher during the adsorption process. The adsorption configurations of the same gas on three MnX-GN catalysts are similar, but the uplift heights of Mn atoms are significantly different. The order of the uplift heights is as follows: MnN_4_-GN > MnDV-GN > MnSV-GN, which agrees with the pioneering works [31].

Moreover, a completed linear fitting between the adsorption energies and the value of the integrated projected COHP (-IpCOHP) is obtained as depicted in Figure 3a–c. Notably, the correlation coefficients are all greater than 0.85, indicating that there is a significant positive correlation between -IpCOHP values and adsorption energies. Therefore, the -IpCOHP values can be used as the descriptor of adsorption strength. The -IpCOHP values of the three gases on MnDV-GN and MnN_4_-GN are lower than those on MnSV-GN, indicating that the two catalysts have lower activity compared to MnSV-GN.

In order to clarify the correlation among the adsorption energies, height and -IpCOHP values of the three gases on MnX-GN, the average height differences of the three gases on MnX-GN are calculated with the values as 0.03 Å, 0.11 Å and 0.35 Å, respectively. From Figure 3d–f, it can be concluded that the -IpCOHP values decrease as the height difference of Mn atoms increase, and it is remarkable that the correlation coefficients between the -IpCOHP values of the three gases and the height of the hump are all greater than 0.9, indicating that there is a significant negative correlation between the -IpCOHP values and the height of the hump. Therefore, the height difference of the Mn atom can be used as the activity descriptor of the adsorption system.

### 3.3. Electronic Structure and Magnetic Properties

The type of gas adsorption will affect the electronic structure distribution of the substrate, including chemical and physical adsorption. We have also investigated the Bader charge and the charge density difference between the gas molecules and MnX-GN (Appendix A and Appendix A), showing that the doped metal atom can enhance the interaction between the reactant and substrate. For example, the charge density difference between the reactive CH_2_O gas and MnX-GN was analyzed, as listed in Appendix A. It was found that the distribution of electrons is mainly concentrated near CH_2_O molecules, indicating the electron acceptor nature of CH_2_O. Similar changes are found in Appendix A for HCN adsorption. However, in Appendix A, the positive charge is mainly distributed around the H_2_S molecule, indicating the electron donor nature, which is similar to the case of FeDV-GN [24]. Moreover, the calculation results of Bader charge are consistent with those of electron density difference. From these results, it can be inferred that the redistribution of charge is mainly caused by the adsorption of gas molecules.

In order to clarify the magnetic properties of gas-adsorbed MnSV-GN, the magnetic moment was analyzed in the process of gas adsorption. By comparing the magnetic moments in Appendix A, we found that the change in magnetic strength for the gas/MnSV-GN system is the biggest (from 3.00 to 1.00 µ_B_). To further discuss the cause of different magnetic properties of the gas/MnX-GN system, the projected crystal overlap Hamilton population (-pCOHP) curves between the gas and the surface were analyzed. The results show that the Mn-C bond in the MnSV-GN structure exhibits anti-bonding features around the Fermi level, which implies a potential electronic instability, as illustrated in Appendix A. When the three gases were adsorbed on the MnSV-GN (Appendix A–d), the interaction between the Mn and C atoms exhibited significant differences from that in Appendix A. The change in magnetic properties for the gas/MnN_4_-GN system is smallest (from 3.05 to 2.90 µ_B_). Moreover, the three gases-adsorbed MnN_4_-GN systems can also be analyzed by -pCOHP curves (see Appendix A). As shown in Appendix A–d, the curves of -pCOHP between Mn and N atoms exhibit nearly no change after gas adsorption. This may indicate that the adsorption of gas molecules has nearly no effect on the magnetic behavior of the substrates. The three gases adsorbed on the MnDV-GN system can also be analyzed by the -pCOHP curve (see Appendix A). There exists a significant difference between the situations in Appendix A on the interaction between the Mn atoms and C atoms. However, there is nearly no difference between Appendix A. The -pCOHP curve analysis is consistent with the magnetic moments in Appendix A. These illustrated that the magnetic property of MnX-GN can be adjusted by gas adsorption. From another point of view, the MnSV-GN catalyst has a stronger activity.

It is observed that the distribution of spin density underwent more remarkable changes between the gas and MnSV-GN support, as depicted in Appendix A. The up spin is dominant on the Mn atom in MnSV-GN, which is opposite for the carbon atom adjacent to Mn. In the case of the three gas-adsorbed MnN4-GN, the spin density is depicted in Appendix A, respectively. The spin electron distribution (the yellow area) of CH_2_O/MnDV-GN composite is mainly concentrated on the O atoms of CH_2_O molecules and Mn atoms (see Appendix A). For H_2_S- and HCN-adsorbed MnDV-GN systems (see Appendix A), the spin density of H_2_S is strongly localized around the Mn atom, and the adsorption of HCN leads to the local polarization of the whole system. In general, the results of the spin density analysis are consistent with the magnetic moment calculation.

### 3.4. Electronic Structure–Reactivity Analysis

In order to explore the relationship between the adsorption effect and the electric structure of MnX-GN catalysts, Fermi softness analyses were performed. Based on the chemical reaction theory, the entire frontier electron band on the solid surface participates in the reaction [62,63], and the electronic states near the Fermi level contribute more to the bonding interaction. Hence, both weight functions (w(E)) and density of states (g(E)) can determine the reactivity of the solid surface. This can quantify the contribution of each electronic state to surface bonding [62]. The Fermi softness (*S_F_*) is defined by the weighted sum of the reactivity contributions, which is the reactivity index of the solid surface. Namely, *S_F_* can be expressed by the formula (Equation (1)):
(1)$$S_{F}=\int_{-\infty }^{+\infty }g\left(E\right)w\left(E\right)dE$$
where g(E) can be assigned from the total density of states, w(E) can be assigned from the derivative of the Fermi–Dirac function, −ƒ′_T_ (*E* − *E_F_*). The −ƒ′_T_ (*E* − *E_F_*) can be expressed by the formula (Equation (2)):
(2)$$-{ƒ}^{\prime }{}_{T}\;(E-E_{F})=\frac{1}{kT}\cdot \frac{1}{\left(e^{\left(E-E_{F}\right)/kT}+1\right)\left(e^{\left(E_{F}-E\right)/kT}+1\right)}$$
where *T* is the parametric temperature, *k* is Boltzmann’s constant and $E_{F}$ is the Fermi level. The w(E) distribution is affected by the value of *kT*, so the Fermi softness is significantly affected by the value of *kT*. Fermi softness is considered to have a quantitative relationship with surface reactivity [62]. Gao et al. extended Fermi softness and applied it to a single atom catalyst system [31]. When *kT* is 1.35 eV, one has also calculated the relationship of *S_F_* and adsorption energy between the four Fe/GS and Hg^0^ [64]. Therefore, it is necessary to select a suitable *kT* value to analyze the activity of the catalyst by Fermi softness.

In order to clarify the effect of *kT* on Fermi softness, the Fermi softness of MnX-GN was calculated at different values of *kT* (Figure 4a). In Figure 4a, the *S_F_* of MnX-GN adsorbents gradually increase and then decrease with the increased kT. Based on the fact that Fermi softness and adsorption energy have a very strong correlation at this appropriate *kT* temperature [10], we have also used the maximum correlation coefficient (R^2^) to find the appropriate *kT* value. In Figure 4b, the magenta, red and blue dotted lines represent the *kT* values of the CH_2_O, H_2_S and HCN, respectively, corresponding to the maximum correlation coefficients. The R^2^ curves of CH_2_O, H_2_S and HCN have the largest peaks at *kT* = 2.8, 1.7 and 1.9 eV, respectively. When *kT* is 2.8 eV, R^2^ has the maximum value of 0.94. At this temperature, the Fermi softness of MnX-GN is around 7.8~8.2 eV. When *kT* is 1.7 and 1.9 eV, the correlation coefficient (R^2^) curves of H_2_S and HCN are all 1. At 1.7 and 1.9 eV temperature, the Fermi softness of MnX-GN for H_2_S and HCN is around 6.4~7.0 and 6.8~7.3 eV, respectively. Therefore, the selected *kT* value is very important for the Fermi softness analysis due to the fact that Fermi softness correlates intimately with the value of kT.

In order to further verify the relationship between the adsorption energy and Fermi softness, the curve of the adsorption energies of CH_2_O, H_2_S and HCN can be plotted as a function of Fermi softness (in Figure 5). From Figure 5, the maximum value of R^2^ under *kT* = 2.8 (1.7, 1.9 eV) is 0.93 (1, 1) and there is a significant negative correlation between Fermi softness and adsorption energy. The higher the adsorption energy is, the larger the Fermi softness. Moreover, Fermi softness can be used as an efficient descriptor for the adsorption analysis of single-atom Mn-doped catalysts.

## 4. Conclusions

In summary, by using the density functional theory, the adsorption properties, including adsorption energy, adsorption geometry, electronic structure and spin distribution, of CH_2_O, H_2_S and HCN are investigated on MnSV-GN substrates with different defects. The order of the uplift height of Mn atoms is as follows: MnN_4_-GN > MnDV-GN > MnSV-GN. The electronic structure of the graphene support determines the adsorption characteristics of CH_2_O, H_2_S and HCN. Among the adsorption energies of the three gases on MnX-GN bases, MnSV-GN is the largest, indicating that the adsorption activity of MnSV-GN is higher than that of MnDV-GN and MnN_4_-GN.

The adsorption energy of the gas/MnX-GN system is linearly related to the uplift height, IpCOHP values and Fermi softness. The variety in anti-bonding and bonding characteristics on every side of the Fermi level in the magnetic -pCOHP curves can be used as the magnetic field strength descriptor of the adsorption system. The adsorbed activity results are consistent with those analyzed by Fermi softness. Properties such as Fermi softness, the uplift height and -IpCOHP can be used as important indicators of adsorption activity when we theoretically design new graphene support materials. Therefore, we hope that the Mn-doped graphene of our research can become an effective resource and provide viable materials for improving the design of graphene-based support.

## Figures and Tables

**Figure 1 molecules-27-02315-f001:**
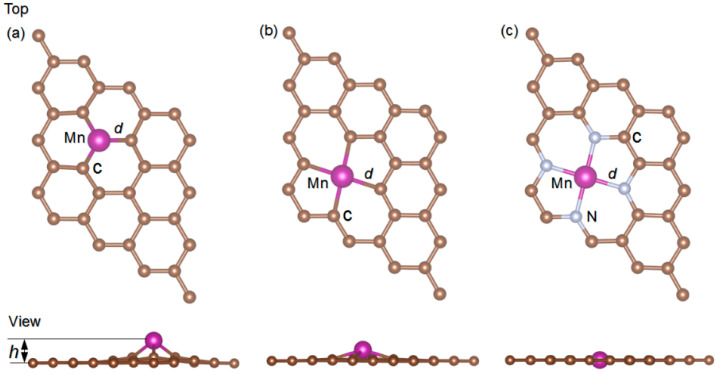
Optimized structures of (**a**) MnSV-GN, (**b**) MnDV-GN and (**c**) MnN_4_-GN, the bond lengths (d, Å), adsorption height of Mn atom (h, Å).

**Figure 2 molecules-27-02315-f002:**
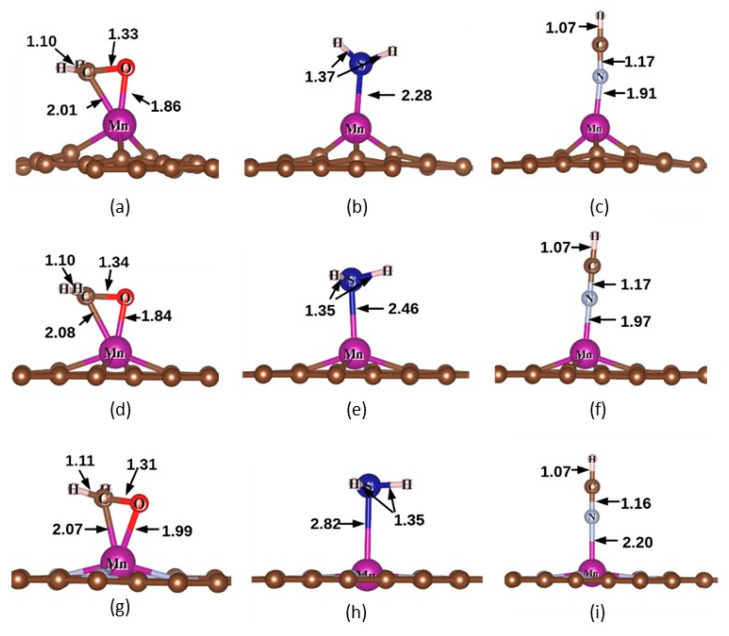
The optimized structures of (**a**) CH_2_O/MnSV-GN, (**b**) H_2_S/MnSV-GN, (**c**) HCN/MnSV-GN, (**e**) CH_2_O/MnDV-GN, (**f**) H_2_S/MnDV-GN, (**g**) HCN/MnDV-GN, (**h**) CH_2_O/MnN_4_-GN, (**i**) H_2_S/ MnN_4_-GN and (**j**) HCN/ MnN_4_-GN. The selected bond distance is expressed in angstroms.

**Figure 3 molecules-27-02315-f003:**
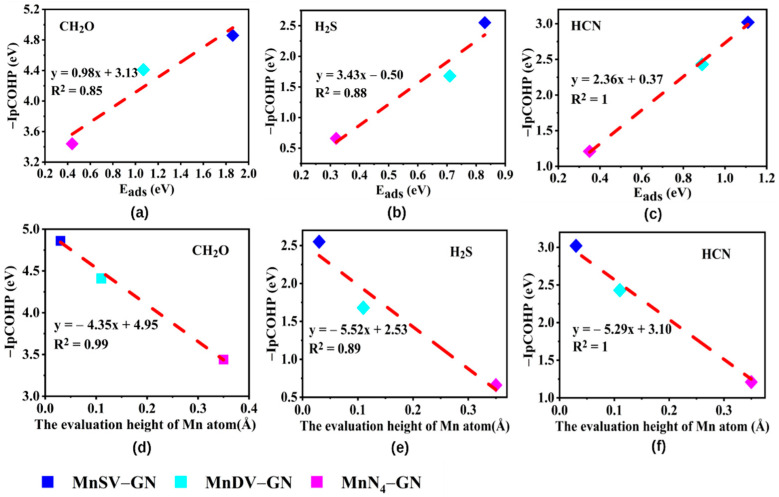
The -IpCOHP values of gases as a function of adsorption energies of (**a**) CH_2_O, (**b**) H_2_S, and (**c**) HCN and Mn atomic heights of (**d**) CH_2_O, (**e**) H_2_S, and (**f**) HCN.

**Figure 4 molecules-27-02315-f004:**
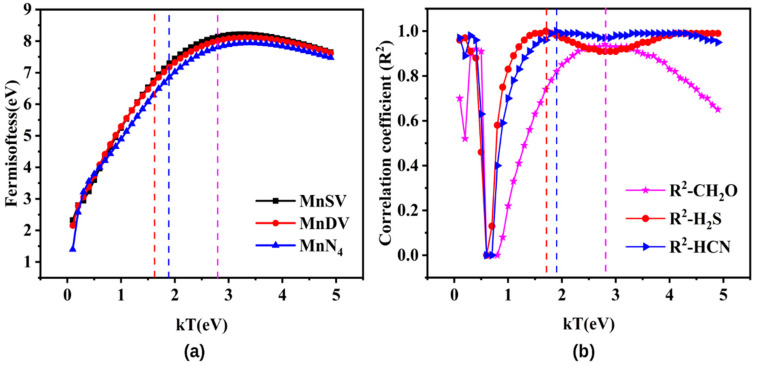
The effect of *kT* on (**a**) Fermi softness of MnX-GN and (**b**) the square correlation coefficient.

**Figure 5 molecules-27-02315-f005:**
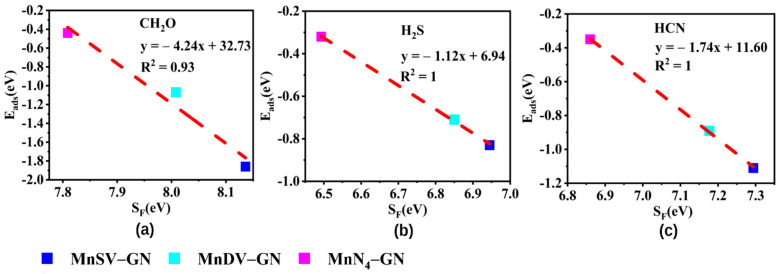
The adsorption energies of (**a**) CH_2_O, (**b**) H_2_S and (**c**) HCN as a function of Fermi softness.

## Data Availability

All data generated or analysed during this study are included in this published article and its Appendix A.

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
