# Peer review of "Adsorption Characteristics of Gas Molecules Adsorbed on Graphene Doped with Mn: A First Principle Study"

_molecules, 2022, doi:10.3390/molecules27072315_

Round 1
Reviewer 1 Report
The authors addressed most of the reviewer’s comments improving the quality of the manuscript, however some flaws still persist.
A further revision of English is required. I have noticed several errors (e.g. missing articles and missing plurals) and also had some difficulties to understand some sentences such as:
Line 209 – “From these results, it can be inferred that the redistribution of charge caused by the chemical adsorption of gas molecules.”
Line 223 –“In Figures S5 (b), (c) and (d), the adsorption properties displayed are all physical adsorption and the interaction between Mn atoms and C atoms is surprisingly weak compared to pristine (Figure S5 (a)).”
Line 291 – “In summary, by using the density functional theory, adsorbed activity and effect of CH2O, H2S and HCN on MnSV-GN with different graphene-based monolayer have been investigated, which are affected by the adsorption energy, adsorption geometry, electronic structure and spin distribution.”
Other minor comments:
Line 253 Where -> where
Line 256 Where -> where
Author Response
The authors addressed most of the reviewer’s comments improving the quality of the manuscript, however some flaws still persist.
A further revision of English is required. I have noticed several errors (e.g. missing articles and missing plurals) and also had some difficulties to understand some sentences such as:
Author reply:
We are sorry for the mistakes and we have corrected the expressions in the manuscript.
Line 209 – “From these results, it can be inferred that the redistribution of charge caused by the chemical adsorption of gas molecules.”
Author reply:
Thank you for reminding and we have corrected the expressions as follows:
From these results, it can be inferred that the redistribution of charge is mainly caused by the adsorption of gas molecules.
Line 223 –“In Figures S5 (b), (c) and (d), the adsorption properties displayed are all physical adsorption and the interaction between Mn atoms and C atoms is surprisingly weak compared to pristine (Figure S5 (a)).”
Author reply:
Thank you for reminding and we have corrected the expressions as follows:
As shown in Figures S4 (b), (c) and (d), the curves of -pCOHP between Mn and N atoms exhibit nearly no change after gas adsorption. This may indicate that the adsorption of gas molecules has nearly no effect on the meganetic behavior of the substrates.
Line 291 – “In summary, by using the density functional theory, adsorbed activity and effect of CH2O, H2S and HCN on MnSV-GN with different graphene-based monolayer have been investigated, which are affected by the adsorption energy, adsorption geometry, electronic structure and spin distribution.”
Author reply:
Thank you for reminding and we have corrected the expressions as follows:
In summary, by using the density functional theory, the adsorption properties, including adsorption energy, adsorption geometry, electronic structure and spin distribution, of CH2O, H2S and HCN are investigated on MnSV-GN substrates with different defects.
Other minor comments:
Line 253 Where -> where
Line 256 Where -> where
Author reply:
Thank you for reminding and we revised it into correct form.

Reviewer 2 Report
The authors investigated the electronic structures, magnetic properties and adsorption energies of Mn doped graphene. The topic is interesting enough but major revisions are needed.
Major
The authors describe their results in terms of adsorption energy, but looking at Figure S2 and its analysis it appears that the molecules are chemisorbed. The author should clarify this point.
This referee belive including Figure S2 in the main text, after Figure 1, could favor the understanding of the whole work.
Minor
pg 3 line 107. In order to improve calculation accuracy and save time simultaneously, a k-point grid with 7 × 7 × 1 as the center is used to sample the Brillouin region for structural optimization". At this point it is not clear why using the k-point grid improves the calculation and saves time. Compared to what?
pg 3 line 144. "inclduding"
pg 5 line 207 "the election donor nature. which is"
pg 6 line 246 "Based on the chemical reaction theory, , the entire frontier".
Author Response
The authors investigated the electronic structures, magnetic properties and adsorption energies of Mn doped graphene. The topic is interesting enough but major revisions are needed.
Major
The authors describe their results in terms of adsorption energy, but looking at Figure S2 and its analysis it appears that the molecules are chemisorbed. The author should clarify this point.
Author reply:
Thank for your good suggestion and we have added the corresponding description in the manuscript.
Besides, the chemisorbed behavior can also be reflected in Figure 2, where the molecules can form chemical bonds with Mn atoms in the substrates.
This referee belive including Figure S2 in the main text, after Figure 1, could favor the understanding of the whole work.
Author reply:
Thank you for the suggestion and we have added Figure S2 into the main text.
Minor
pg 3 line 107. In order to improve calculation accuracy and save time simultaneously, a k-point grid with 7 × 7 × 1 as the center is used to sample the Brillouin region for structural optimization". At this point it is not clear why using the k-point grid improves the calculation and saves time. Compared to what?
Author reply:
Thank for your good suggestion. We have performed a comprehensive test for the k-point grid. As shown in figure below, it can be seen that the calculation time is the least at the kpoint of 7 x 7 x 1, from which the change of energies tends to be quasi stable.
pg 3 line 144. "inclduding"
pg 5 line 207 "the election donor nature. which is"
pg 6 line 246 "Based on the chemical reaction theory, , the entire frontier".
Author reply:
Thank you for reminding and we have revised it into correct form.

Round 2
Reviewer 1 Report
The issues raised earlier seem essentially to have been resolved and the paper is now suitable for publication.
Reviewer 2 Report
The authors responded to all requests.
This manuscript is a resubmission of an earlier submission. The following is a list of the peer review reports and author responses from that submission.
Round 1
Reviewer 1 Report
The authors investigated the electronic structures, magnetic properties and adsorption energies of Mn doped graphene.
The topic is interesting enough but the manuscript is unclear, major revisions are needed. While reading, this reviewer noticed that the description of the figures and tables in some cases does not match, making it impossible to understand the text.
Example:
Page 3. "The electronic properties of MnSV-GN, MnDV-GN and MnN4
-GN are showed in Table S1."
Table S1 reports also geometric data, so it results not very clear to the reader.
Pag 4. "The optimized configurations and parameters of adsorption systems are shown in Figure 2."
Figure 2 shows "The -IpCOHP values of gases as a function of adsorption energies and Mn atomic heights". The optimized configurations seems to be reported in Figure1.
Page 4. "In order to investigate the influence of gas adsorption on MnX-GN catalysts, adsorption energies of three gas/MnX-GN catalysts are calculated, as listed in Table S2"
This reviewer is not able to undestand how is organized Table S2. It is not understandable.
Page 4. "In addition, the adsorption of gases on MnSV-GN can exhibit chemical adsorption (see Table S2)."
It is not clear, waht is the criteria?
Page 4. ".. the adsorption energies and the IpCOHP value is obtained as depicted in Figure 3".
Maybe the authors refers to the Figure 2.
Page 5. ".. in Figures 4(a)-(j)"
Actually in figure 4 there are just 3 panels A, B and C.
Page 5-7. Figure 5, Figure 6, Figure 7 and Figure 8 do not exist.
These are a few examples, but they are sufficient to state that the article in this form is complex to judge.
Other major revisions.
The use of "-pCOHP" in the abstract is not clear. Since it is used as a main descriptor, a better definition of this quantity is needed
The authors evalaute magnetic properies, so they have to declare if they used spin-polarized DFT scheme to tkae in account open-shell electronic configurations.
Author Response
Comments and Suggestions for Authors
The authors investigated the electronic structures, magnetic properties and adsorption energies of Mn doped graphene.
The topic is interesting enough but the manuscript is unclear, major revisions are needed. While reading, this reviewer noticed that the description of the figures and tables in some cases does not match, making it impossible to understand the text.
Response: Many thank the encouraged comments. We have improved below suggestions point to point in the new version.
Example:
Page 3. "The electronic properties of MnSV-GN, MnDV-GN and MnN4
-GN are showed in Table S1."
Table S1 reports also geometric data, so it results not very clear to the reader.
Response: The electronic properties displayed in Table S1 mainly include the charge transfer and magnetic moments. We are sorry for the unclear description and we have added the illustrations in detail in the manuscript.
Pag 4. "The optimized configurations and parameters of adsorption systems are shown in Figure 2."
Figure 2 shows "The -IpCOHP values of gases as a function of adsorption energies and Mn atomic heights". The optimized configuration seems to be reported in Figure1.
Response: Thank you for reminding and we have realized the mistake and revised it in the manuscript.
Page 4. "In order to investigate the influence of gas adsorption on MnX-GN catalysts, adsorption energies of three gas/MnX-GN catalysts are calculated, as listed in Table S2"
This reviewer is not able to undestand how is organized Table S2. It is not understandable.
Response: We have revised Table S2 into a more clear formation and substituted data between “-IpCOHP” and “Δh”.
Page 4. "In addition, the adsorption of gases on MnSV-GN can exhibit chemical adsorption (see Table S2)."
It is not clear, What is the criteria?
Response: We judged the adsorption behavior via the adsorption energies ( Eads>0.5 eV belong to the chemical adsorption), which are big enough for gas adsorption on MnSV-GN.
Page 4. ".. the adsorption energies and the IpCOHP value is obtained as depicted in Figure 3".
Maybe the authors refer to the Figure 2.
Response: Sorry for the mistake and we have revised it in the manuscript.
Page 5. ".. in Figures 4(a)-(j)"
Actually in figure 4 there are just 3 panels A, B and C.
Response: Sorry for the mistake, the charge density difference is displayed in Figure S3. We have revised it in the manuscript.
Page 5-7. Figure 5, Figure 6, Figure 7 and Figure 8 do not exist.
Response: Thanks for pointing out this mistake. In the new manuscript, we have modified these.
These are a few examples, but they are sufficient to state that the article in this form is complex to judge.
Other major revisions.
The use of "-pCOHP" in the abstract is not clear. Since it is used as a main descriptor, a better definition of this quantity is needed
Response: Thanks for the reminding about not giving the definition of “-pCOHP” and we have add the corresponding explanation for the “-IpCOHP” and “-pCOHP” in the abstract.
The authors evaluate magnetic properties, so they have to declare if they used spin-polarized DFT scheme to take in account open-shell electronic configurations.
Response: Thank you for the comment. Actually, we have applied the spin-polarized DFT to calculate magnetic properties and now give a description in the part 2 “Computational details and methods”
Reviewer 2 Report
The manuscript entitled “Adsorption Characteristics of Gas Molecules Adsorbed on Graphene Doped with Mn: A First Principle Study” presents a study addressing the interactions of a collection of small molecules with Mn-doped graphene substrates by means of DFT calculation techniques.
I do see some relevance in the subject of the study, but a thorough revision is necessary. In order to be published, several aspects have to be clarified and a general improvement has to be achieved. The most important points, as well as some minor corrections, are:
Apparently, the authors didn’t include van der Waals interactions in this study or didn’t mention its inclusion. For correctly describing the interaction of the molecules with the surface, van der Waals interactions should have been considered. Furthermore, the Vasp code allows inclusion of dispersive interactions with different approaches and different computational costs that should have been taken into account.
Considering the expressions for the binding and adsorption energies, contrary to what is stated in the manuscript, positive values resulting from these expressions represent energetically favourable adsorption, while negative values represent non favourable interactions. In Point 3.1 the authors refer negative binding energies and consider it as favourable interactions with the surface. In Point 3.2 the authors report positive adsorption energies and discuss it as favourable interactions. This point must be fully clarified.
The authors try to extract correlations between some properties like the IpCOHP vs adsorption energy or IpCOHP vs lifting of the Mn atom from the surface plane, or between adsorption energies of CH2O, H2S, and HCN and Fermi softness by performing linear fittings between these pairs with only 3 points. Of course, the significance of these poor statistics is highly questionable, and no profound conclusions can be inferred, or should be taken with great care.
Although the figures present enough quality, there are throughout the manuscript several references to figures that don't belong to the manuscript but are included in the Supplementary Info instead, ex Figure 4 instead of Figure S4 or Figure 6 instead of Figure S6, etc. Also, when the authors refer to Figures 7(a) and 7(b) they should refer to Figures 3(a) and 3(b).
The authors write that “In order to improve the accuracy of calculation and save time simultaneously, a k-point grid with 7 × 7 × 1 as the center is used to sample the Brillouin region for structural optimization. When the energy change is less than 10 meV/atom, the k-point grid test is finished”. It is not clear from the manuscript what was done in practice: several k-points grids have been tested and finally the 7 x 7 x 1 grid has been adopted? Please clarify.
Finally, one of the critical shortcomings is the poor quality of the English writing. The text needs careful language revision.
Following, I remark some additional points that the authors should revise:
- In Subsection2, when discussing adsorption energies, the text should make reference to table S2.
- The legend of Figure 2 should include the meaning of the points colours.
- Although “COHP” is correctly described in the text as “crystal orbital Hamilton population”, there is no description in the text of what the acronyms “Ip-COHP” or “p-COHP” stand for.
- In Figure 1 there is no description or reference to the meaning and values attributed to the letters d and h.
Author Response
Comments and Suggestions for Authors
The manuscript entitled “Adsorption Characteristics of Gas Molecules Adsorbed on Graphene Doped with Mn: A First Principle Study” presents a study addressing the interactions of a collection of small molecules with Mn-doped graphene substrates by means of DFT calculation techniques.
I do see some relevance in the subject of the study, but a thorough revision is necessary. In order to be published, several aspects have to be clarified and a general improvement has to be achieved. The most important points, as well as some minor corrections, are:
Response: Many thanks for the confirmation and encouraged comments. We have improved below suggestions point to point in the new version.
Apparently, the authors didn’t include van der Waals interactions in this study or didn’t mention its inclusion. For correctly describing the interaction of the molecules with the surface, van der Waals interactions should have been considered. Furthermore, the Vasp code allows inclusion of dispersive interactions with different approaches and different computational costs that should have been taken into account.
Response: Many thanks for the very helpful comments and give a description in the part 2 “Computational details and methods”.
Considering the expressions for the binding and adsorption energies, contrary to what is stated in the manuscript, positive values resulting from these expressions represent energetically favourable adsorption, while negative values represent non favourable interactions. In Point 3.1 the authors refer negative binding energies and consider it as favourable interactions with the surface. In Point 3.2 the authors report positive adsorption energies and discuss it as favourable interactions. This point must be fully clarified.
Response: We have realized the contraries between different parts. We have revised the expression of adsorption energy as Eads = Egas-Mn/GN – EMn/GN – Egas. Thus, negative values indicate energetically favorable adsorption. We have also revised the numerical values into negative form in Point 3.2.
The authors try to extract correlations between some properties like the IpCOHP vs adsorption energy or IpCOHP vs lifting of the Mn atom from the surface plane, or between adsorption energies of CH2O, H2S, and HCN and Fermi softness by performing linear fittings between these pairs with only 3 points. Of course, the significance of these poor statistics is highly questionable, and no profound conclusions can be inferred, or should be taken with great care.
Response: Thanks for your good suggestion. In this manuscript, three kinds of graphene substrates were adopted to calculate the correlation between some properties like the IpCOHP vs adsorption energy or IpCOHP vs lifting of the Mn atom from the surface plane. So there are only 3 points in Figure 2 and Figure 4. We have added the illustration about three point’s colours in Figure 2 and Figure 4.
Although the figures present enough quality, there are throughout the manuscript several references to figures that don't belong to the manuscript but are included in the Supplementary Info instead, ex Figure 4 instead of Figure S4 or Figure 6 instead of Figure S6, etc. Also, when the authors refer to Figures 7(a) and 7(b) they should refer to Figures 3(a) and 3(b).
Response: Sorry for the mistake and we have revised it in the manuscript.
The authors write that “In order to improve the accuracy of calculation and save time simultaneously, a k-point grid with 7 × 7 × 1 as the center is used to sample the Brillouin region for structural optimization. When the energy change is less than 10 meV/atom, the k-point grid test is finished”. It is not clear from the manuscript what was done in practice: several k-points grids have been tested and finally the 7 x 7 x 1 grid has been adopted? Please clarify.
Response: Thanks for your good suggestion. Yes, your understand is consistent with our opinion. As shown in figure below, it can be seen that the calculation time is the least at the Kpoint 7 x 7 x 1, from which the change of energies tends to be quasi stable.
Finally, one of the critical shortcomings is the poor quality of the English writing. The text needs careful language revision.
Response: Thanks a lot for your comment. We have revised some sentences in our new version.
Following, I remark some additional points that the authors should revise:
- In Subsection2, when discussing adsorption energies, the text should make reference to table S2.
Response: Thank you for the suggestion and we have emphasized the values of adsorption energies and labeled “see Table S2” in the discussions.
- The legend of Figure 2 should include the meaning of the points colours.
Response: Thank you for the suggestion and we have added the illustration about three point’s colours in Figure 2 and Figure 4.
- Although “COHP” is correctly described in the text as “crystal orbital Hamilton population”, there is no description in the text of what the acronyms “Ip-COHP” or “p-COHP” stand for.
Response: Thank you for your reminding us about giving the definition of “-pCOHP” and “-IpCOHP”. We have add the corresponding explanation for the “-IpCOHP” and “-pCOHP” in the abstract and in the manuscript.
- In Figure 1 there is no description or reference to the meaning and values attributed to the letters d and h.
Response: Thank you for the suggestion and we have added the corresponding illustration in the manuscript.
Round 2
Reviewer 1 Report
The authors have improved the manuscript, but several and important inaccuracies still remain. I would encourage the authors to be more careful in presenting their work.
For example, at pg 2 line 96 phrase: "Because the chemical bond between gas and surface of MnX-GN catalysts play a key role in the adsorption. And without the van der Waals (vdw) correction, gas adsorption on metal [..]" is not clear.
Actually, gas adorption is sensitive to vdW correction and the statement "the chemical bond between gas and surface" dos not make any sense in the framwork of the adsorption process.
Is the investigated gas chemosorbed or adsorbed?
An english revision is need.
Reviewer 2 Report
Although the manuscript has been improved in some parts in accordance with the suggestions of the referees, a general improvement has not been achieved and the manuscript in its current state is unacceptable for publication.
The main objections presented by the referees have only been partially considered or not clarified. For example, the expression for the binding energy is still incorrect.
Also, the authors confirmed they didn’t include van der Waals interactions in their study, which is an essential component for a correct description of the interaction between the molecules and the surface, particularly for those more loosely bound.
Finally, an English revision of the text has not been performed as had been advised.